# McCune–Albright Syndrome: A Case Report and Review of Literature

**DOI:** 10.3390/ijms24108464

**Published:** 2023-05-09

**Authors:** Nicolas C. Nicolaides, Maria Kontou, Ioannis-Anargyros Vasilakis, Maria Binou, Evangelia Lykopoulou, Christina Kanaka-Gantenbein

**Affiliations:** 1First Department of Pediatrics, National and Kapodistrian University of Athens Medical School, “Aghia Sophia” Children’s Hospital, 11527 Athens, Greece; kontou_mar@yahoo.gr (M.K.); vasilakisioan@yahoo.gr (I.-A.V.); mmpinou@hotmail.com (M.B.); lilialykopoulou@gmail.com (E.L.); chriskan@med.uoa.gr (C.K.-G.); 2Reference Center for Rare Pediatric Endocrine Disorders, First Department of Pediatrics, National and Kapodistrian University of Athens Medical School, “Aghia Sophia” Children’s Hospital, 11527 Athens, Greece

**Keywords:** café au lait macules, gonadotropin-independent precocious puberty, growth hormone excess, McCune–Albright syndrome, ovarian cyst, vaginal bleeding

## Abstract

McCune–Albright syndrome (MAS) is a rare sporadic condition defined by the classic triad of fibrous dysplasia of bone, café au lait skin macules, and hyperfunctioning endocrinopathies. The molecular basis of MAS has been ascribed to the post-zygotic somatic gain-of-function mutations in the *GNAS* gene, which encodes the alpha subunit of G proteins, leading to constitutive activation of several G Protein-Coupled Receptors (GPCRs). The co-occurrence of two of the above-mentioned cardinal clinical manifestations sets the diagnosis at the clinical level. In this case report, we describe a 27-month-old girl who presented with gonadotropin-independent precocious puberty secondary to an estrogen-secreting ovarian cyst, a café au lait skin macule and growth hormone, and prolactin excess, and we provide an updated review of the scientific literature on the clinical features, diagnostic work-up, and therapeutic management of MAS.

## 1. Introduction

McCune–Albright syndrome (MAS) is a rare, complex non-hereditable genetic disorder, which is characterized by a variable combination of fibrous dysplasia of bone, café au lait pigmented skin lesions, and hyperfunctioning endocrinopathies [1]. This syndrome was first reported in 1936 by Dr. Donovan J. McCune who described a case of a nine-year-old girl with precocious puberty, multiple sites of skin pigmentation and hyperthyroidism [2]. One year later, Dr. Fuller Albright and collaborators published a series of five female cases presenting with precocious puberty, endocrine dysfunction, several sites of skin pigmentation, and osteitis fibrosa disseminata [3]. In 1942, Lichtenstein and Jaffe used, for the first time, the term “fibrous dysplasia of bone” to describe the bone lesions accompanying the clinical manifestations of MAS [4]. They also clarified that fibrous dysplasia of bone was different from the osseous lesions of neurofibromatosis [5]. Several decades later, Weinstein et al. identified activating mutations in the *GNAS* gene (Guanine Nucleotide-Binding Protein, Alpha-Stimulating Activity Polypeptide), which encodes the alpha subunit of G proteins, in affected tissues from all the patients of the study, as well as in tissues not typically involved in the syndrome [6]. Interestingly, the authors found a significant variation of the proportion of cells that were affected in different tissues, indicating the important role of mosaicism in the pathogenesis of MAS [6]. Since then, the tremendous progress of genetics and developmental biology enhanced our understanding of the molecular basis of MAS, clarifying that a post-zygotic substitution of arginine by a cysteine or histidine residue (R201C/H) at codon 201 in exon 8 of *GNAS* is responsible for the clinical manifestations of the syndrome in the majority of the cases (over 95%) [7]. Rarely, patients with MAS may harbor a mutation at codon 227 in exon 9 (Q227R/L) [8]. All these activating somatic mutations have been identified as disease-causing because they lead to constitutive activation of a large number of G Protein-Coupled Receptors (GPCRs), including the LH, FSH, TSH, GHRH, and ACTH receptors [1]. Although GPCRs have been traditionally considered ordered protein structures, a recent study by Fonin and collaborators showed that the GPCR–G protein signaling system is an example of an intrinsic protein disorder [9], since their conformational plasticity and structural flexibility do not comply with the classic dogma of structural biology of “lock and key” model [10].

MAS is a rare sporadic disease with an estimated prevalence between 1/100,000 and 1/1,000,000 [11]. The cardinal clinical manifestations of the syndrome include the classic triad of monostotic/polyostotic fibrous dysplasia, café au lait skin pigmentation, and several hyperfunctioning endocrinopathies, such as gonadotropin-independent precocious puberty, growth hormone excess, non-autoimmune hyperthyroidism, hyperprolactinemia, or neonatal hypercortisolism [12]. At least two of the above-mentioned clinical features must be present to set the diagnosis of MAS. Among them, the most severe clinical manifestation is fibrous dysplasia, which may be isolated or in combination with other features of the syndrome, and usually affects the craniofacial bones, femur, and pelvic bones [13]. The bone lesions in MAS are caused by impaired differentiation of skeletal stem cells due to constitutive Gαs activation, thereby leading to imbalanced bone remodeling [14]. Indeed, the histological characteristics of bone lesions with fibrous dysplasia include areas of woven bone with unmineralized osteoid, hypercellular fibrous stroma, collagen fibers termed as “Sharpey fibers”, abundant osteoclasts, and increased vascularity [14,15]. Affected bones display a homogenous “ground glass” appearance on radiographs [16]. In addition, the improper replacement of normal bone by a weak abnormal fibrotic lesion in patients with MAS results in the overproduction of FGF23, which, in turn, decreases 1-α-hydroxylase activity and increases urinary phosphate excretion at the level of the proximal renal tubule [17]. However, hypophosphatemia only develops in patients with MAS who exhibit severe skeletal disease [18].

The therapeutic management of patients with MAS is only symptomatic and requires a multidisciplinary team with experience in pediatrics, endocrinology, orthopedics, pain management, and rehabilitation [19]. Girls with gonadotropin-independent precocious puberty are treated with aromatase inhibitors (letrozole), estrogen receptor modulators (tamoxifen), pure estrogen receptor antagonists (fulvestrant), and anti-androgens [1,12,13]. The treatment of gonadotropin-independent precocious puberty in boys includes aromatase inhibitors (letrozole), androgen receptor blockers (spironolactone, flutamide or cyproterone acetate), and inhibitors of steroidogenesis (ketonazole) [1,12,13]. The non-autoimmune hyperthyroidism usually responds to thionamides (methimazole) for a short-time period; however, most patients prefer thyroidectomy in high-volume surgical centers, as a definitive treatment [1,12,13]. Growth hormone excess is typically well-responsive to somatostatin analogues (octreotide, lanreotide) or growth hormone receptor antagonists (pegvisomant), used either alone or in combination [1,12,13]. Neonatal hypercortisolism is treated with metyrapone as first-line management due to its low risk of hepatotoxicity; however, ketoconazole and mitotane may also be administered before adrenalectomy [1,12,13]. Last, but not least, fibrous dysplasia may be managed with analgesic therapy and intravenous bisphosphonates. The latter should be administered in cases of fibrous dysplasia-related pain, taking into consideration the lowest effective dose and interval [20]. In addition to traditional therapeutic management, cumulative evidence suggests that novel therapies might ameliorate the severe clinical manifestations of MAS (reviewed in [21]). Indeed, denosumab (inhibitor of receptor activator of nuclear kappa-B ligand (RANKL)) and burosumab (monoclonal antibody to FGF23) have been effectively added to the therapeutic armamentarium of MAS [22,23,24,25,26]. However, further and larger studies are still needed to prove the promising benefits. From the molecular point of view, since MAS is a mosaic disorder and there is no intrinsically disordered definition in this pathologic condition, a holistic and specific targeted therapy has not been achieved so far.

In this case report, we describe a 27-month-old Greek girl with vaginal bleeding due to an ovarian cyst of the right ovary leading to gonadotropin-independent precocious puberty, a café au lait skin macule, and growth hormone excess, indicative of the diagnosis of MAS. This study was reviewed and approved by the “Aghia Sophia” Children’s Hospital Committee on the Ethics of Human Research, approval number 1707-01/02/2022. Written informed consent was obtained from the parents of the patient for the publication of this case report and any accompanying images. All data generated or analyzed during this case report are included.

## 2. Case Report

A Greek girl presented at the age of 27 months with vaginal bleeding lasting three days. Mucous vaginal discharge without signs of bleeding had been noted seven days before. The girl was born after normal delivery at full term and was the first child of non-consanguineous parents. She weighed 3550 gr (50th centile) with a length of 50 cm (50th–75th centile) at birth and had no gross congenital anomalies. There was no past medical history and no history of medications. The mother had developed gestational diabetes which was managed with dietary measures.

Upon admission at the hospital, an abdominal ultrasound was performed which showed an ovarian cyst of the right ovary measuring 5 cm × 4.1 cm × 3.5 cm with clear content and a thin layer. The uterus was found to be of the infantile type with a thickened secretory endometrium (of 1 cm thickness) and a thickened vagina. No adrenal pathology was found.

The girl was initially admitted to the pediatric surgery department for 2 days. Laboratory investigations revealed increased levels of estradiol (111.9 pg/mL), prolactin (79.94 ng/mL (normal range: 5–20 ng/mL)) and growth hormone (26.5 ng/mL) with both random blood FSH and LH levels ˂ 0.1 mUI/mL and normal IGF-1 (122 ng/mL (18–172 ng/mL)).

The girl was subsequently transferred to the pediatric department. Careful systemic clinical evaluation revealed a café-au-lait skin macule with jagged borders measuring 2 cm × 2.5 cm on the lower back which respected the midline. Coarse facial features and hyperpigmented labia minora were also noted. The child’s weight, height, and head circumference were along the 90th–97th centile, 50th–75th centile, and 75th–90th centile, respectively (Figure 1). Breast development was rated as stage II according to Tanner staging with nipple hyperpigmentation, and pubic and axillary hair were rated as stage I. The bone age was assessed and found to be normal at 2 years and 9 months (6 months advanced in relation to chronological age).

A further endocrine work-up was then performed. The levels of testosterone, dehydroepiandrosterone sulfate (DHEAS), and androstenedione were found to be within normal reference range: testosterone ˂20 ng/dL, androstenedione: ˂0.3 ng/mL, DHEAS: ˂15 μg/dL. CA 125, CA 19-9, and β-HCG were also within normal range: CA 125: 15.4 U/mL (normal range ˂ 35 U/mL), CA 19-9: 31.5 U/mL (normal range ˂ 34 U/mL), and β-HCG: 0.1 mIU/mL). Due to the clinical and laboratory indications of GH-excess, an oral glucose tolerance test was performed which demonstrated no suppression of GH (nadir GH: 5.67 ng/mL at 120 min). Subsequently, a brain and pituitary MRI was performed which revealed no abnormalities.

The adrenal evaluation showed normal ACTH (44.99 pg/mL) and cortisol (21.90 μg/dL) concentrations. Measurements of TSH (3.25 μUI/mL) and fT4 (1.19 ng/dL) concentrations were within normal range, and the thyroid ultrasound depicted no abnormal findings.

The MRI of the lower abdomen and pelvis revealed a cystic lesion measuring 3.94 × 3.85 × 2.26 cm in diameter without features of a benign or malignant tumor.

A cardiac echocardiogram was performed and showed no pathology.

Based on the clinical and laboratory presentation of gonadotropin-independent estradiol production resulting in precocious puberty, the ovarian cyst, the laboratory evidence of growth-hormone excess along with the presence of the café-au-lait skin macule with characteristic features of irregular borders (Coast of Maine) and a distribution showing the so-called “respect of” the midline of the body, a clinical diagnosis of McCune–Albright Syndrome was made. The girl was then evaluated for additional features of McCune–Albright Syndrome with an assessment of the skeletal system: a thorough skeletal survey was performed (including all four limbs and the pelvis) which showed no signs of fibrous dysplasia.

The girl was then discharged with a plan for follow-up by a multidisciplinary team consisting of Pediatricians, Endocrinologists and Orthopedics. Due to evidence that bone lesions may develop over time, the skeletal assessment was recommended to be repeated in the future. However, shortly after discharge, due to the recurrence of vaginal bleeding and parental anxiety, the decision was made to start treatment of precocious puberty with an aromatase inhibitor. At that time, the patient did not receive any medication for growth hormone and prolactin hypersecretion. As the family moved away from Greece, further follow-up was planned to occur abroad in another pediatric endocrine department.

## 3. Discussion

In our case report, the 27-month-old girl presented with a 3-day history of painless vaginal bleeding and was first admitted to the pediatric surgery department in “Aghia Sophia” Children’s Hospital, Athens, Greece. Upon admission, a physical examination revealed no signs of sexual abuse, trauma, or foreign body, while the initial laboratory investigations did not show thrombocytopenia, coagulation disorders, or infection. The abdominal ultrasound depicted a large ovarian cyst of the right ovary with clear content and a thin layer, excluding the possibility of a benign or malignant tumor. Moreover, CA 125 and CA 19-9 measurements were within a normal range, further excluding malignancy from the differential diagnosis of vaginal bleeding. Importantly, endocrinologic evaluation revealed high estradiol concentrations with both suppressed FSH and LH, indicating that the ovarian cyst was autonomously estrogen-producing, and suggesting gonadotropin-independent precocious puberty (Figure 2). Indeed, a detailed physical examination upon admission to the general pediatric clinic showed breast development II according to the Tanner staging. In addition, we identified a café-au-lait skin macule with jagged borders on the lower back which respected the midline. Both these features of café au lait skin pigmentation and the presence of gonadotropin-independent precocious puberty pointed to a diagnosis of MAS (Figure 2). To further prove hormonal hypersecretion, and due to her coarse facial characteristics, the patient underwent an oral glucose tolerance test that revealed the absence of suppression of growth hormone secretion. Growth hormone excess was autonomous since a brain and pituitary MRI did not show any adenoma (Figure 2). Finally, fibrous dysplasia was not identified during the thorough skeletal survey; however, the majority of bone lesions usually develop later in life before the age of 15 years [16,27,28]. Hart and collaborators demonstrated that the majority of affected bones are identified between the age of 3 and 10 years [28]; therefore, fibrous dysplasia might develop later in our patient.

The diagnosis of MAS was set clinically in this girl. The presence of a café-au-lait skin macule with irregular borders (Coast of Maine) and a distribution respective of the midline of the body, in association with gonadotropin-independent precocious puberty and growth hormone excess fulfilled two of the cardinal criteria of MAS diagnosis [29]. We did not send a blood sample for identification of any *GNAS* mutation(s) because the mutation detection rates in PCR-based sequencing methods are approximately 20–30% in peripheral blood lymphocytes and >80% in affected tissues [30,31,32] and the parents were reluctant to perform further investigations in their daughter. Moreover, a negative result would not rule out MAS diagnosis. The two most important factors influencing the detection of *GNAS* mutations are the level of mosaicism in the tested tissue and the sensitivity of the method [1,29]. Intense efforts have been made towards the improvement of identification of *GNAS* mutations in patients with MAS [33,34,35,36]; however, the ideal method has not been identified yet.

The patient was first admitted to the pediatric surgery department in “Aghia Sophia” Children’s Hospital. The detailed physical examination (café-au-lait skin spot and breast development II according to Tanner staging), as well as the initial endocrinologic evaluation (gonadotropin-independent precocious puberty secondary to autonomous estrogen-secreting ovarian cyst), were indicative of the diagnosis of MAS preventing any surgical management. Papanikolaou and Michala have reviewed the published literature about current management for autonomous ovarian cysts in prepubertal girls and concluded that a conservative approach should always be the first-line choice, taking into consideration the high risk of recurrence in cases of MAS [37]. Undoubtedly, the presence of an ovarian cyst requires a multidisciplinary approach with a close collaboration of pediatricians, endocrinologists, surgeons, and pediatric gynecologists to prevent unnecessary cystectomy or oophorectomy. In the past, a small number of cases of MAS who underwent “mistaken identity” oophorectomy has been published, thereby emphasizing the importance of timely MAS diagnosis through fruitful collaboration among different specialties [38,39].

The co-occurrence of gonadotropin-independent precocious puberty and growth hormone excess in our patient is rare in cases with MAS [40]. Both the diagnosis and therapeutic management of growth hormone excess are challenging, since not only growth hormone excess but also precocious puberty is characterized by increased growth velocity and advanced bone age [38]. Recently, Zhai and co-workers published a case series of patients with MAS with growth hormone excess and precocious puberty [40]. They found that the coexistence of growth hormone excess and precocious puberty in MAS is more common in girls than in boys. Furthermore, these patients displayed increased growth velocity Z-scores and advanced bone age; however, the early and appropriate control of both conditions resulted in reduced growth velocity and a more stabilized bone age [40]. Based on these important findings, the regular follow up of these patients is *sine qua non* to achieve the best therapeutic outcomes.

## 4. Conclusions

In summary, we described a 27-month-old Greek girl with gonadotropin-independent precocious puberty secondary to an estrogen-producing ovarian cyst, a café au lait skin macule and growth hormone excess, all suggestive of MAS diagnosis and highlight the importance of careful assessment and diagnostic work-out in order to prevent unnecessary or even dangerous surgical approach.

## Figures and Tables

**Figure 1 ijms-24-08464-f001:**
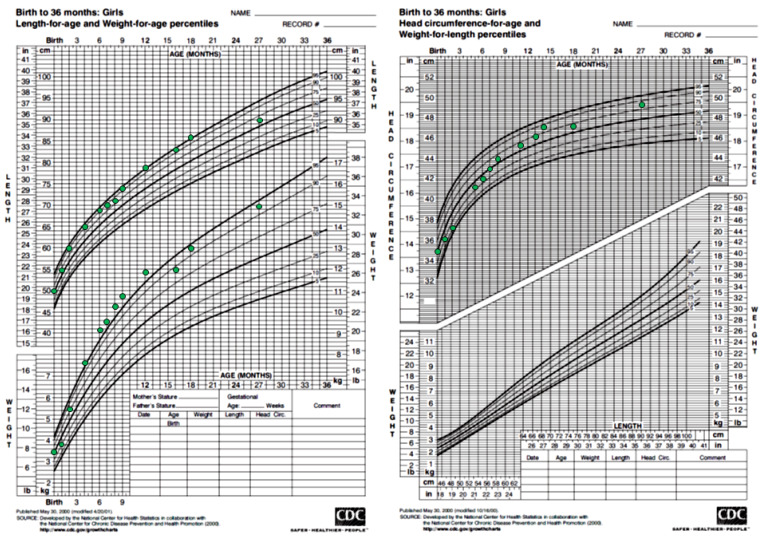
Auxological parameters of the patient.

**Figure 2 ijms-24-08464-f002:**
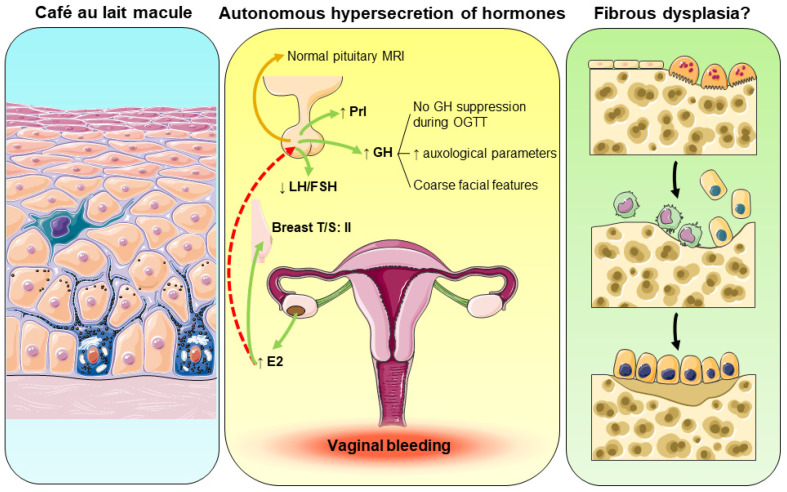
Pathophysiologic aspects of the reported case. ↑: increase; ↓: decrease.

## Data Availability

Data sharing not applicable.

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
