# Peer review of "McCune–Albright Syndrome: A Case Report and Review of Literature"

_ijms, 2023, doi:10.3390/ijms24108464_

Round 1

Reviewer 1 Report

The authors report a case report and review of literature for MAS. The G-proteins are studied as ordered proteins structures but recently it was found that these proteins are disordered. Please see

Coskuner-Weber et al., Biopyhysical Reviews, 2022, 14, 679. 

Regarding the literature review, the authors should describe the lack of intrinsically disordered definition in MAS and how it may effect treatment design. 

Author Response

We thank the Reviewer for this comment. We have added two short comments on these novel findings of structural biology of GPCRs as intrinsically disordered proteins (page 2, highlighted in yellow color; page 3, highlighted in yellow color).

Reviewer 2 Report

Nicolaides and coworkers report a case of a 27-month-old girl in whom a clinical diagnosis of McCune Albright syndrome(MAS) was made on the basis of the occurrence of gonadotropin-independent precocious puberty secondary to an estrogen-secreting ovarian cyst, a cafe au lait skin macule and GH and Prolactin hypersecretion without alterations of pituitary imaging on MRI. They moreover provide a review of scientific literature on the various aspects of this syndrome.

COMMENT

Major concerns:     

MAS is a rare disorder characterized by the  co-occurrence of the classic triad of fibrous dysplasia, cafe au lait skin macules and hyperfunctioning endocrinopaties. The molecular basis  of this syndrome  has been ascribed to post-zygotic somatic, gain-of-function mutations  in GNAS gene, leading to mosaic Galphas activator and inappropriate production of intracellular cAMP.     The location and extent of affected tissues are responsible for the clinical phenotype.      The presence of at least two  manifestations of the  classic triad  in this case (lack of fibrous dysplasia) could allow a clinical diagnosis , however,, data on the molecular pattern of this patient should be very useful. Concerning this, the authors affirm" we did not send a blood sample for identification of any GNAS mutations, because the mutation detection rates in PCR-based sequencing methods are approximately 20-30% in peripheral blood lymphocytes and > 80% in peripheral tissues". In my opinion, this does not justify the failure to carry out the  molecular study and I suggest to perform it, in order to give the pathophysiological seal to the diagnosis of this case.                                                                                                                                                                                                           Minor concerns:

- How do the authors explain the discrepancy between the high GH levels and the normal IGF1 levels?                                                                                  - The girl was treated  with an aromatase inhibitor for recurrence of vaginal bleeding but the authors do not clarify whether the GH  and PRL hypersecretion was treated ,even if in discussing the literature data in Introduction section, they  described the therapeutic management of these conditions.

Author Response

We thank the Reviewer for the first comment; however, our decision not to send a blood sample from the patient for identification of any GNAS mutation was based both on the reluctancy of the parents to allow any further investigations to be performed in their daughter and since, as we stated in the manuscript, “a negative result would not rule out MAS diagnosis”, “because the mutation detection rates in PCR-based sequencing methods are approximately 20-30% in peripheral blood lymphocytes and > 80% in peripheral tissues”. The diagnosis of MAS remains clinical and is based on the co-occurrence of at least two of the classic triad of fibrous dysplasia, cafe au lait skin macules and hyperfunctioning endocrinopaties.

We agree with the Reviewer’s second comment. In most studies of MAS, levels of IGF-1 were elevated. Very rarely, levels of IGF-1 were in the normal range [please see references 8,17,19 in “McCune-Albright Syndrome in Infant with Growth Hormone Excess” by Brzica K et al. Genes (Basel). 2022 Jul 27;13(8):1345]. In addition, several studies supported that GH may act directly on the growth plate in an IGF-1-independent fashion [please see references 23,24,25,26,27 in “McCune-Albright Syndrome in Infant with Growth Hormone Excess” by Brzica K et al. Genes (Basel). 2022 Jul 27;13(8):1345].                                              

We thank the Reviewer for the third comment. We have added a sentence stating that “The patient did not receive any medication for growth hormone and prolactin hypersecretion as the family moved away from Greece and further follow-up was planned to occur abroad” (page 5, highlighted in yellow color).

Round 2

Reviewer 2 Report

I believe the paper has been improved also following the reviewers' comments and suggestions. Moreover the authors gave an acceptable justification for not performig the molecular study.